# BlockTicket: A framework for electronic tickets based on smart contract

Amjad Aldweesh [ID] *

Affiliation College of Computer Science and IT, Shaqra University, Shaqra, Saudi Arabia

* a.aldweesh@su.edu.sa

**Data Availability Statement:** All relevant data are within the paper.

**Funding:** The authors received no specific funding for this work.

## Abstract

As the use of digital subscription services like electronic tickets (E-ticketing) has grown in the age of e-commerce, so too have instances of copyright and violation. Because it is dependent on the centralized authority administration of authoritative institutions, the traditional E-ticketing system has a significant cost associated with it. Blockchain, which is a distributed system, has the characteristics of decentralization, anonymity, auditability, security, and persistency. These attributes allow it to address the problems that are currently being experienced by the E-ticketing system. In this study, we present a framework for E-ticketing that makes use of blockchain technology. The blockchain-based electronic ticketing model eliminates the involvement of third parties while also lowering the potential of data leaks and improving users' levels of privacy. This is accomplished by separating the credential information of users from the financial transactions. In the meanwhile, a blockchain implementation of the existing E-ticketing architecture has the potential to improve throughput, reduce the amount of redundant work, and boost the efficiency of consensus. An examination of the experimental data shows that the framework has a number of advantages, some of which are a high throughput, flexible scalability, and efficient ticket holding times.

## Introduction

E-commerce service providers have evolved in the Web 2.0 era to facilitate the sale of digital goods, including e-books, e-music, and e-tickets. These two-sided marketplaces allow content producers to market their products to Buyers online. So, the platform is a take for content producers and Buyers, providing both benefits. When it comes to payments, the platform handles the nuts and bolts on the back end for the creative. The platform's content delivery network (CDN) facilitates the transfer of digital goods from the seller to the customer.

In many cases, the platform will also handle authorization on the vendor's behalf, using (Digital Right Management) DRM and other technical means to ensure compliance. In addition, the e-commerce hub handles promotion for the content producer. Buyers benefit from centralized platforms because they make it easier to find and access the E-tickets of several vendors. Current methods include using product search engines and recommender systems to get buyers the items they need. In various ways, blocking consequences result from the centralized nature of the platform's operations. There is a blocking impact on both the content developer

**Competing interests:** The authors have declared that no competing interests exist.

and the consumer. The data they collect is the foundation for most of the value that centralized e-commerce platforms offer to businesses. Consumers' platform usage and spending habits are tracked in minute detail.

Machine learning algorithm take this information as input in order to benefit both the content producer and the buyer by making more targeted product recommendations, for instance. This boosts revenue for the content producer. The customer saves time in their search thanks to the tip. The result is a blocking effect for content producers that is driven by data. Many platforms use the blocking effect to their advantage by prohibiting the export of acquired data in the event that authors wish to switch to a competing platform. As a result, all training must be restarted on the new system. There is a blocking of pricing power when platforms like the Amazon Marketplace or Apple's music streaming service dictate the prices at which digital material can be purchased. The platform's reliance on proprietary DRM solutions has a restrictive effect on the consumer. Most of them use their own proprietary file format to store your newly purchased e-books or music. Therefore, users can only get to them through a locked-down app.

Since users cannot simply transfer their previously held tickets from one platform's DRM to another, this makes it more difficult for consumers to switch to another platform. This results in a blocking impact on the data format. Channel blocking effects have evolved in e-commerce platform solutions as the popularity of subscription-based services has grown in recent years. Channel blocking occurs, for example, when a user subscribes to a service that only offers their preferred type of tickets, such as football season tickets.

We suggest the decentralization of various value propositions as a means of combating lock-in effects that can occur in online commerce involving E-tickets. For this reason, we developed BlockTicket, a blockchain-based application for managing E-tickets.

To sum up, the contribution of this article is proposing BlockTicket, a blockchain-based E-ticket management framework, to address the blocking effects of centralized e-tickets platforms on content providers and users. BlockTicket provides an open, transparent alternative to centralized e-tickets platforms. This article describes BlockTicket's architecture, roles, and functions as well as past work on decentralizing digital content as well as e-tickets. BlockTicket's technological and economic features are also critically examined and compared to centralized e-tickets platforms. The proposed technique addresses the lock-in consequences of centralized e-tickets platforms in an innovative and promising way.

The remaining of this paper are broken down as follows: In Section II, an overview of previous work toward a more decentralized digital content and blockchain as the technical building block for BlockTicket are presented. The topic is broken down in further detail in Section III. The role model, the architecture of the system, and the various functions that are associated with the various roles are all described. In Section IV, we will demonstrate an example of a prototypical implementation of the suggested approach. In Section V, we conduct a critical analysis of the approach. Both the technical and the economic issues are shown. In this section, we compare the methods used by decentralized platforms with centralized ones to trade digital material. In Section VI, this study comes to an end and finishes with some observations on the necessary additional work.

## Preliminary

### Blockchain

All transactions in a blockchain are recorded in a series of connected blocks that are encrypted for safety. This creates a robust connection between blocks, ensuring their sequential order and providing an implicit, robust timestamp method. The result is that it is impossible to

change a block without also affecting all of its descendants. A distributed database, or shared ledger, can be constructed from the information stored in a blockchain [1]. Blockchain's distinctive qualities as a functioning scheme include its immutability, timestamping, and the elimination of the need to trust a central authority.

Permissionless blockchains and permissioned blockchains are the two main architectural styles for blockchains. The public blockchain, which Bitcoin and Ethereum belong to, allows anybody with internet access to view and update transaction records in real time (i.e., there is no membership requirement). Permission blockchains [2] are different since they are exclusive to a select group of people. A new block is confirmed by the network once it has been added to the chain according to a predetermined protocol. Various blockchains use different consensus protocols, each with their own unique implementation details (such as proof of work or POW). For instance, a consensus in a public blockchain may take the shape of a hash puzzle, the solution to which is locating a fixed hash value.

However, the increased security this consensus process provides to the chain comes at the expense of computational power and time (it can resist up to 50% rogue nodes). For example, Bitcoin can only handle 7 transactions per second at most, and obtaining final consensus can take up to an hour. Permission blockchains, on the other hand, use a voting-based Byzantine fault-tolerant algorithm as its consensus mechanism, such as Practical Byzantine Fault Tolerance (PBFT) or Stellar Consensus Protocol (SCP), which avoids the need for computationally expensive hash puzzles. This results in a larger transaction throughput because consensus may be reached more quickly. However, permissioned blockchains typically necessitate the trustworthiness of more than two-thirds of nodes, as opposed to 51%. Consensus algorithms are discussed in greater depth in [3].

## Electronic ticketing

E-ticketing systems were divided by Vives-Guasch et al. [4] into those that used smart cards and those that didn't. Ticketing services are restricted in the smart-card-based systems [5], which rely on contacting and contact-less smart cards as the medium. For systems that don't rely on smart cards, the user interface is typically a mobile phone, which is capable of a wider range of tasks than a smart card. This research focuses on the second sort of scheme, the ticket system. Issuer, user, collector, and a third-party supplier [4, 6, 7] are the usual players in an e-ticketing system. The contents of a ticket are determined by the issuer, who then makes and distributes the ticket to the buyer. The user presents the ticket to the collector, who then transfers the items or provides the service to the user in exchange for payment. A ticket may be transferred between users, and the original purchaser and holder need not be different people.

A trustworthy third party is required to verify the issuer, user, and collector's transaction in order to prevent fraud. The third party could be the platform's host or it could just be in charge of verifying trades. Hu et al. [6] suggested Uni-ticket, a unified electronic ticketing platform that supports several ticketing formats. The Uni-ticket system connects with both individual and corporate user ends, and it is built on a reliable third-party platform. Individuals can make queries and reservations via their smartphones, while businesses can distribute and manage tickets online. Every deal is handled by the master server and recorded in the master database. The system as a whole will be rendered inoperable if the main database goes down. Further, dishonest actors can accomplish their fraud goals by penetrating the system's central database and altering the data there.

There is a lack of mutual trust in the ticket system, so a rigorous authentication procedure is necessary to ensure that ticket holders are who they claim to be. Multiple sources of

verification for tickets ensures that individuals in need receive actual services. People can attend events with tickets issued by the government or non-profit organizations.

Given the complexity of the system and the number of entities involved, the question of how to foster trust among them is crucial. In the past, fraudulent agro-dealers had delayed payments to legitimate ones because they had defrauded the government's monetary system [8]. The administrative procedure is also slow because the authorized list must be shared among numerous industries. There is a problem with the e-ticketing system in that beneficiary lists are often submitted late, delaying the delivery and activation of e-cards [9].

## Related work

This section provides a summary and analysis of existing blockchain-based solutions for E-tickets and Digital-contents, as well as an overview of the relevant work in this area.

There are a few distinct ways that distributed ledger technology have been put into use for decentralized e-commerce, e-government and e-contents. Distributed Hash Tables (DHTs) and the Inter Planetary File System (IPFS) [10] on a peer-to-peer network are used by Open Bazaar (www.openbazaar.com), a decentralized access web, to store and deliver products to a searching user. Payments can be made with Bitcoin or Zcash thanks to the framework's cryptocurrency support. Unlike other systems, Open Bazaar does not make modular distinctions between jobs. It is instead strongly integrated by a decentralized application. Prasad et al. [11] proposed utilizing the Ethereum blockchain to record purchasers' bids and automate the auctioning and payment processing in an eBay-styled auction use case. IPFS also serves as a repository for critical data and files. When using Beaver [12], customers can shop in complete secrecy. The strategy utilizes a payment system and a reputation system to operate. Non-interactive zero-knowledge proofs [13] and linkable ring signatures [14] are used in a secret-keeping manner to connect the entities supplying a reputation assertion to the statement itself. This strategy treats customers and sellers in a different way. More nuanced role distinctions are not made.

Access between users and a public pool of critical data is made easier with the blockchain-based paradigm presented in [15]. The basic goal of the approach was to provide a safe means of sharing data while keeping its confidentiality intact. There was a lack of clarity regarding the protocols and algorithms required for inter-entity communication and authentication. In order to protect patients' confidentiality, the authors of [16] proposed a blockchain-based system for exchanging health records. The authors used cloud storage and blockchain indexes to get around the centralized data storage issue. To protect users' anonymity when exchanging data, we employed a signature technique for extracting relevant information and an attribute-based access control approach. In addition, smart contracts were outlined to specify permissions and provide secure data access. The authors of [17] suggest a system for the safe exchange of PHI that is based on the blockchain. An enhanced capacity for medical diagnosis inspired the development of this paradigm. Specifically, private and consortium blockchains are used in the work suggested. The former is employed for the actual storage of PHI, while the latter is employed for the maintenance of indexes to such data. Protected health information (PHI) and patient identities are secured with a keyword search method employing public key encryption. When applied to medical diagnostics, the provided methodology guaranteed enhancement. However, we don't utilize a system that looks for conjunctive keywords.

Authors in [18] developed a blockchain-based paradigm for safe data sharing in V2V networks (VANETs). The suggested framework is referred to as a consortium blockchain-based data storage and sharing system. Digital signatures and bilinear pairing methods are used to guarantee the authenticity and consistency of the data in the proposed model. Vehicles that

take part in data sharing are rewarded with "data coins" for reliable data transfer. With the proposed model, securely sharing and storing data was guaranteed. However, verification of the cars' authenticity is ignored. On order to ensure the safety of data exchanged between vehicles in a network, the authors of [19] implemented a consortium blockchain model. A reputation-based methodology is employed to ensure that only high-quality data is shared. The reputation system described uses a three-weight subjective approach for management. Security analysis is also carried out in the planned task, which helps to spread safety and trust.

A blockchain-based paradigm for cloud service providers to share data was proposed by the authors of [20]. The proposed architecture made use of smart contracts and an access control policy to efficiently track the actions of data owners and revoke access in the event of rule and permission violation. With the suggested approach, cloud service providers could safely accomplish data provenance and auditing, and users could safely share medical data without worrying about the privacy of their data.

In [21], authors presented a data-sharing platform that integrates IPFS, Ethereum's blockchain, and ABE to secure and share data. The model's primary goal was to enable secure, granular data access management. Finally, the issue of the cloud server not delivering the correct search results was fixed, and a keyword search feature was added to the cipher text of the distributed storage system. The system lacked, however, the definition of the mechanism by which user permission revocation to update access policy might be implemented. In [22], the authors suggested a system for mobile crowd-sensing that combines blockchain technology with deep reinforcement learning for secure data gathering and sharing. Distributed ledger technology (blockchain) was implemented to allow for data sharing amongst mobile terminals with varying levels of security.

In [17], the authors recommend using a system called MedBlock to manage all of the patients' personal information. To implement the proposed concept, blockchain and distributed ledger technology were utilized. The suggested approach used a consensus mechanism to decrease both energy use and network congestion. When it comes to securely transmitting sensitive medical information, the authors state that MedBlock was a crucial component.

In [23] order to address these concerns, the developers of Privacy-Preserving Auction Scheme [24] suggested a solution (PPAS). Two separate entities—the auctioneer and the intermediate platform—form the third party in the proposed concept. In the suggested paradigm, an auction is conducted utilizing homomorphic encryption. An improved approach, Enhanced PPAS, is presented to further strengthen security (EPPAS). The robustness of the two proposed methods is evaluated against a variety of attacks. To ensure the practicability of the proposed methods, extensive performance evaluations were conducted. To ensure the safety of Lightweight Clients, the authors of [25] devised a secure service provisioning approach (LCs). Using blockchain technology, the network is protected and users' personal information is kept private.

For e-government, real estate and e-cities, blockchain has been deployed to solve different issues. The real estate industry is invaluable to a country's economy and society [26]. Registering property and assets is one of the emerging uses of Blockchain technology in e-governance [27]. Smart contracts on the blockchain allow for instant, verifiable transactions between parties. More and more people in the research and development communities are interested in designing and coding frameworks and infrastructures for land registry systems that utilize Blockchain technology [28]. These safe and open systems support the transfer of ownership and mortgage registration. Land records kept with Blockchain technology are more trustworthy and reliable, contributing to the growth of national wealth [29]. The real estate industry is notoriously slow, but blockchain technology may help speed things up by replacing time-consuming human processes. There is no need to stress over document duplication, or loss [30].

Blockchain technology can significantly lower the cost of asset registries since it eliminates the need for middlemen [31]. More importantly, Blockchain technology can be used to detect and halt unlawful or shadow real estate transactions [32].

Recent years have seen the introduction of blockchain technology in E-Ticket systems, which brings with it the benefits of decentralization, security, and trust. To stop "ticket theft from posted photographs" from disclosing vital information about tickets, the author of [33] examined E-Ticket systems in a narrow context and proposed a blockchain-based solution. Using a smart contract, we were able to implement the ticketing system. However, the organization retains discretion over the final ticket status. Using Ethereum and smart contracts, the author of [44] developed a protocol for blockchain-based E-Ticket systems, and implemented a blockchain-based BTS for electronic ticketing.

In addition, the authors of [34, 35] investigate the use of blockchain technology and cloud hosting in an electronic ticketing system. The author modified the previous blockchain protocol in [36]. The upgraded Rollerchain uses the sharding technique and PBFT algorithm, resulting in a blockchain protocol with high efficiency, slow cubical dilatation, small capacity expansion, and good scalability; this protocol is called pruneable sharding-based blockchain. With these alterations, blockchain may become even more useful to us. In addition, in [37, 38], the authors analyzed a variety of topics based on characteristics of the e-ticketing system. Invaluable advancements have been made in the computerized ticketing system thanks to each of these works.

The paper [39] addresses the knowledge acquisition bottleneck in traditional expert systems by incorporating the artificial neural network theory into its development. The paper offers a realized electric operation ticket expert system based on back propagation network. A type of automated ticket sales and distribution system is analyzed in [40].

In addition, [41] investigates reliable NFC ticketing. However, questions about users' privacy still need answers. Authors in [42] discuss the use of blockchain technology and digital wallets to manage e-prescriptions in a secure and efficient manner. The authors propose a system that ensures the privacy of patient data while preventing double-spending, which is a common problem in the healthcare industry. The proposed system uses smart contracts and permissioned blockchain to enable secure transactions and ensure the integrity of the data. The authors conclude that blockchain-based e-prescription management can improve the efficiency and security of healthcare systems, but further research and development are needed to overcome the technical and regulatory challenges. With the goal of assuring adherence to COVID-19 health protocols, [43] offers an enhanced design for a multi-sport event ticketing accounting information system that includes RFID and blockchain technology. The suggested solution aims to improve the accuracy, efficiency, and transparency of ticket sales and management while avoiding physical contact among stakeholders, limiting virus propagation, and protecting the system against fraud. However, all those paper used permissioned ledger.

Even though a significant amount of research has been done on both E-ticket system and digital content, none of that research has investigated preventing blockage of the E-ticketing system using the public blockchain technology. In addition, there are several limitations to the related works discussed. For instance, there is a lack of clarity regarding the protocols and algorithms required for inter-entity communication and authentication, as well as the mechanism by which user permission revocation to update access policy might be implemented. Verification of the authenticity of the cars is ignored in some cases. Some systems do not make nuanced role distinctions between customers and sellers, and more importantly, these solutions do not seem to have been widely adopted. Table 1 present a comparison of our proposed framework to recent frameworks published between 2020 to 2022.

**Table 1. Comparison of the proposed framework and the existing work between 2020 and 2022.**

| Publication | Targeted Problem | Framework | Implementation | Stakeholders |
|---|---|---|---|---|
| [44], | E-tickets for events | Yes | No | User, Node and Merchants. |
| [45], | Prices on E-Ticket selling | No | No | Seller/Byers. |
| [46], | E-vouchers System | Yes | Yes | Non-profit organization, the beneficiary, and the dealer. |
| [47], | E-Ticket for Transportation | No | Yes | User, Admin and Transportation Org. |
| [48], | Event E-Tickets | Yes | No | Ticket issuers, Visitor and Event organizers |
| BlockTicket | Suit for all E-tickets types | Yes | Yes | User, Issuers, Distributors and Auditors. |

## Methodology

The development of a blockchain-based E-ticketing framework is a complex and demanding process that requires extensive preparation and research. To achieve this, a systematic and rigorous approach was undertaken, consisting of the following measures:

Firstly, a clear definition of the problem was established to determine the desired outcomes of the blockchain-based E-ticketing framework. This enabled the focus of the project to be narrowed down, and the parameters of the project to be defined more precisely.

Secondly, an analysis was conducted to identify the key stakeholders who are likely to benefit or be negatively impacted by the blockchain-based E-ticketing framework, including attendees, event planners, and ticket vendors. Their perspectives, wants, and concerns were taken into consideration in the design process.

Thirdly, an assessment of the current state of the art was performed to gain an understanding of the functioning of existing E-ticketing frameworks and to identify any potential limitations or issues. This allowed for improvements to be made to the current methods.

Fourthly, a comprehensive review of the requirements for the blockchain-based E-ticketing framework was conducted, based on research and discussions with relevant parties. Key considerations included safety, scalability, price, and convenience.

Fifthly, a model of the blockchain-based E-ticketing framework was created based on the identified needs. This involved the development of a design paper and a working prototype utilizing blockchain technology.

Sixth, as part of the iterative process, prototypes were tested and feedback from stakeholders was gathered. Criticism and feedback were incorporated into future iterations of the blockchain-based E-ticketing framework's architecture. Finally, upon completion of the E-ticketing framework built on the blockchain, procedures for regular maintenance and any necessary updates were implemented in order to ensure optimal performance and longevity.

In summary, the blockchain-based E-ticketing framework was made using a strict and organized process. This process included a thorough analysis of key stakeholders and their needs, as well as the use of current state-of-the-art technologies and iterative design processes to make sure the end product was effective and efficient.

We adhere to this technique to make sure that our blockchain-based E-ticketing architecture is well-researched, well-designed, and suitable for all parties involved.

## BlockTicket proposed framework

In this section, we go over the specifics of our framework, which is based on the Ethereum blockchain and is designed to validate the delivery of E-tickets. This solution enables the secure delivery of E-tickets without relying on a central authority, as well as automatic payment and resolution of any disputes that may arise. Blockchain technology, Ethereum smart contracts,

## Ticket Generation and Distribution

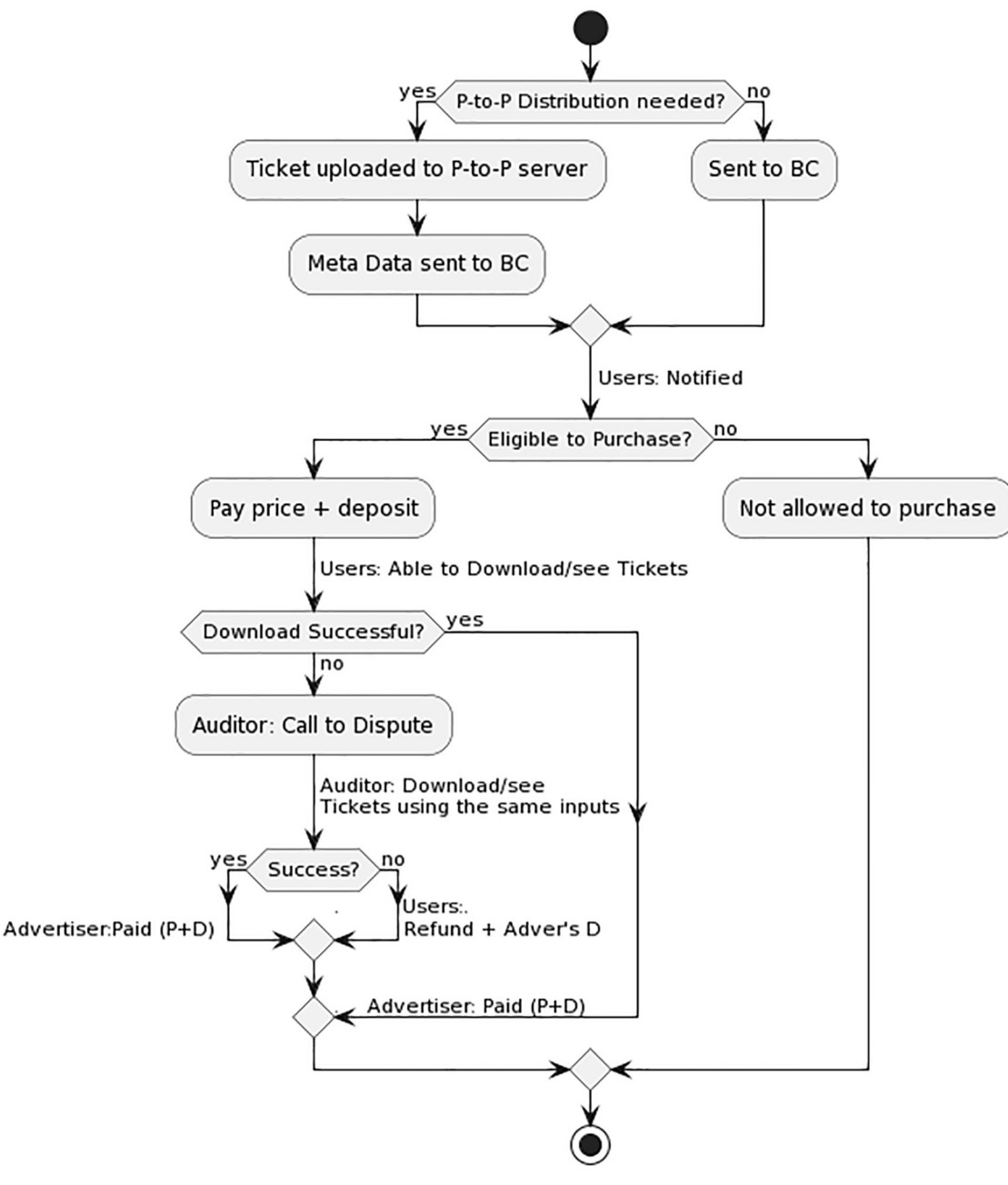

**Fig 1. BlockTicket flow diagram.**

and optionally distributed Peer-to-Peer files system are all integral parts of this solution. Fig 1 depicts the flow diagram of the proposed framework.

The proof of the transmission of E-tickets between two parties is the primary focus of the suggested blockchain system. All the entities involved have Ethereum addresses and take part in the conversation between the blockchain, the smart contract and in some cases with distributed Peer-to-Peer files system. Also, all parties involved initially sign a formal agreement form.

As such, the formed contract includes a hash of the terms and conditions agreement stored in advertisers' side, such as, the Inter Planetary File System (IPFS) [49]. IPFS is a peer-to-peer, distributed file system that can efficiently store massive amounts of data. As a result, the blockchain is only utilized to keep the meta data that the IPFS offers for a saved E-ticket on the advertisers' sides [50] and if and only if the size of the E-ticket manageable on the blockchain, as storing chunks of data there is too expensive. If both parties are in agreement, the transaction will proceed, and the advertiser and the buyer will be debited the agreed-upon deposit amounts. The roles in the proposed framework include:

**E-tickets:** that would own and/or offered for sale by buyers. **Buyers:** one or multiple users who can own E-tickets by making a request via the smart contract. Buyers who accept terms and conditions, they can make a request for E-tickets in exchange for the cost of that ticket and a deposit that will be held as an incentive for trustworthiness. After that, **Advertiser:**, which could be one or multiple, who collect tickets and offer them for sale. would put up the same amount of deposit. Next, the contract will immediately provide the buyer a fresh digital-currencies. **Digital-currencies:** the user might then use this digital-currencies to safely download their tickets off-chain (i.e., advertiser side). A successful transaction requires that the buyer check their ticket before the digital-currencies expires. When a buyer successfully retrieves the ticket from a advertisers' sides, they will notify the smart contract. Once the buyer confirms the transfer was successful, the transaction is complete and payment is made. As soon as the **Tickets issuers:**, who is a company or person who has legal possession of E-tickets, and the advertisers' are compensated, the deposit is restored to all parties. In the event of a disagreement, the **Auditor:**, who confidence in the Auditor is shared by E-tickets issuers, advertisers, and buyers. would attempt to check the same ticket as buyers by exchanging their digital-currencies for the buyers'. A reimbursement could happen depending on the auditor's conclusion. Fig 2 depicts the entities and the blockchain interactions with one another.

## Implementation and evaluation

### Implementation

Using the Remix IDE, the smart contract is developed and then put through its tests. An environment for building smart contracts in Solidity [51], as well as an environment for testing and debugging the written contracts, is made available by Remix. The details of the implementation and the testing are the primary emphases of this section.

It all starts with a ticket issuer informing the advertisers about an event or anything that requires a ticket to attend or watch. Next, advertisers execute a function on the smart contract to announce that event as well as its terms and conditions. Then, a buyer requests to buy an E-ticket through the smart contract. As soon as a buyer submits a request, it is assumed that they have read and accepted the terms and conditions of the contract, and that they have already deposited the required deposit. The same sum would be deposited by the advertiser as stated above.

This ensures that all entities involved have the same amount of power and serves as an incentive for everyone to be honest. After the buyer makes a request for a certain ticket, the contract will generate a digital-currencies for them automatically. The advertiser would then make it possible for the buyer to use their digital currencies for a safe download/view of that ticket. The advertiser would then trigger a smart contract function to verify the download/view. When a buyer receives this message, it means they have successfully downloaded their ticket from the advertisers' side. With the download confirmation result, the buyer could then trigger a function to confirm.

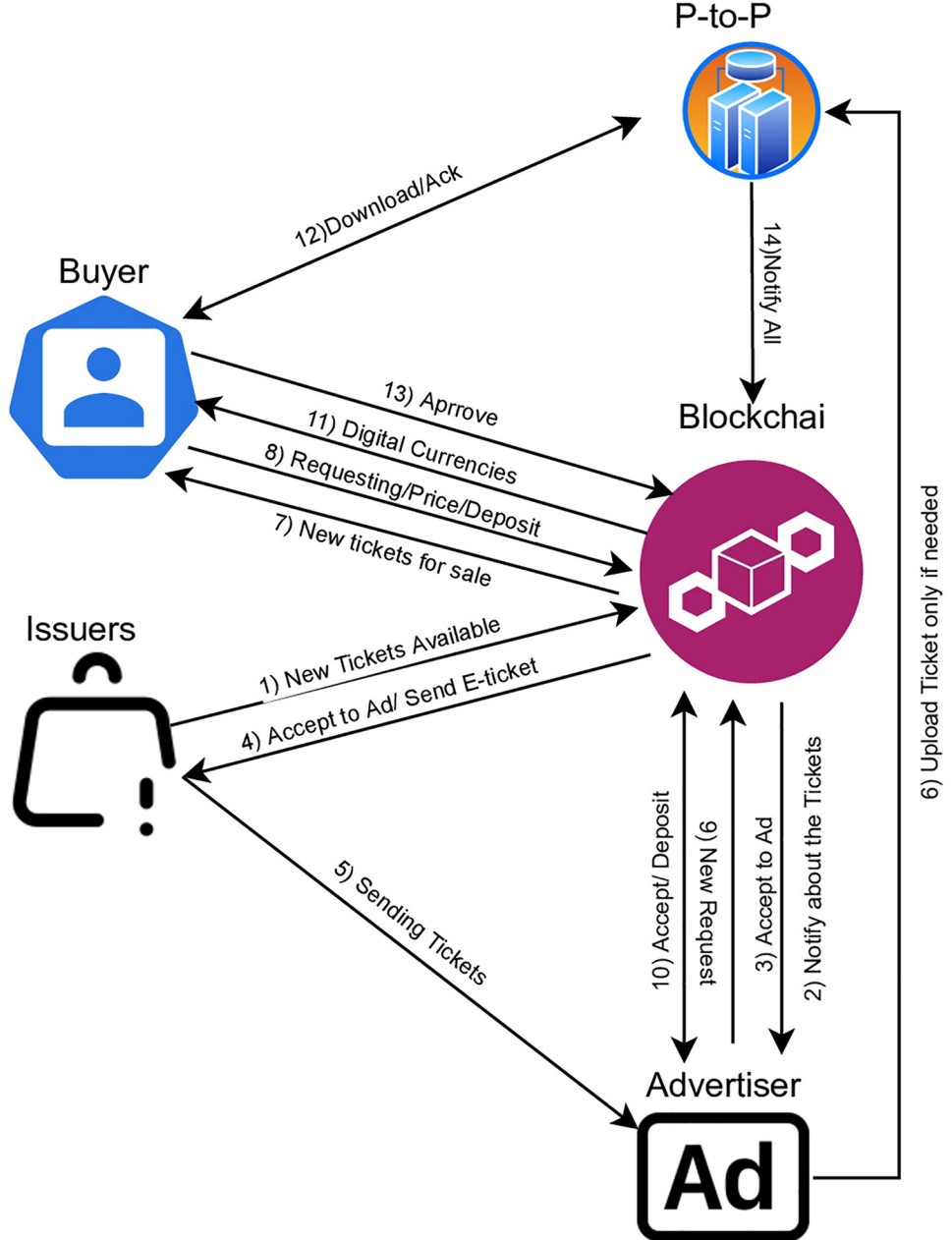

**Fig 2. BlockTicket main entities and their interaction with the blockchain.**

In the event that the buyer is content, the transaction is finalized, and the deposit is returned. The cost of the item would be split between the ticket issuer and the advertiser. If the buyer is not, however, the Auditor will use the buyer's digital currencies to view or download that same ticket. Whether or whether the buyer is eligible for a refund depends on the decision of the Auditor. If the advertiser has not yet deposited its deposit, the buyer can request a refund through the system.

Fig 3 illustrates steps that take place in each of the scenarios described above (i.e., Success, Unsuccess and Discontent), respectively. The flow chart presents the whole procedures,

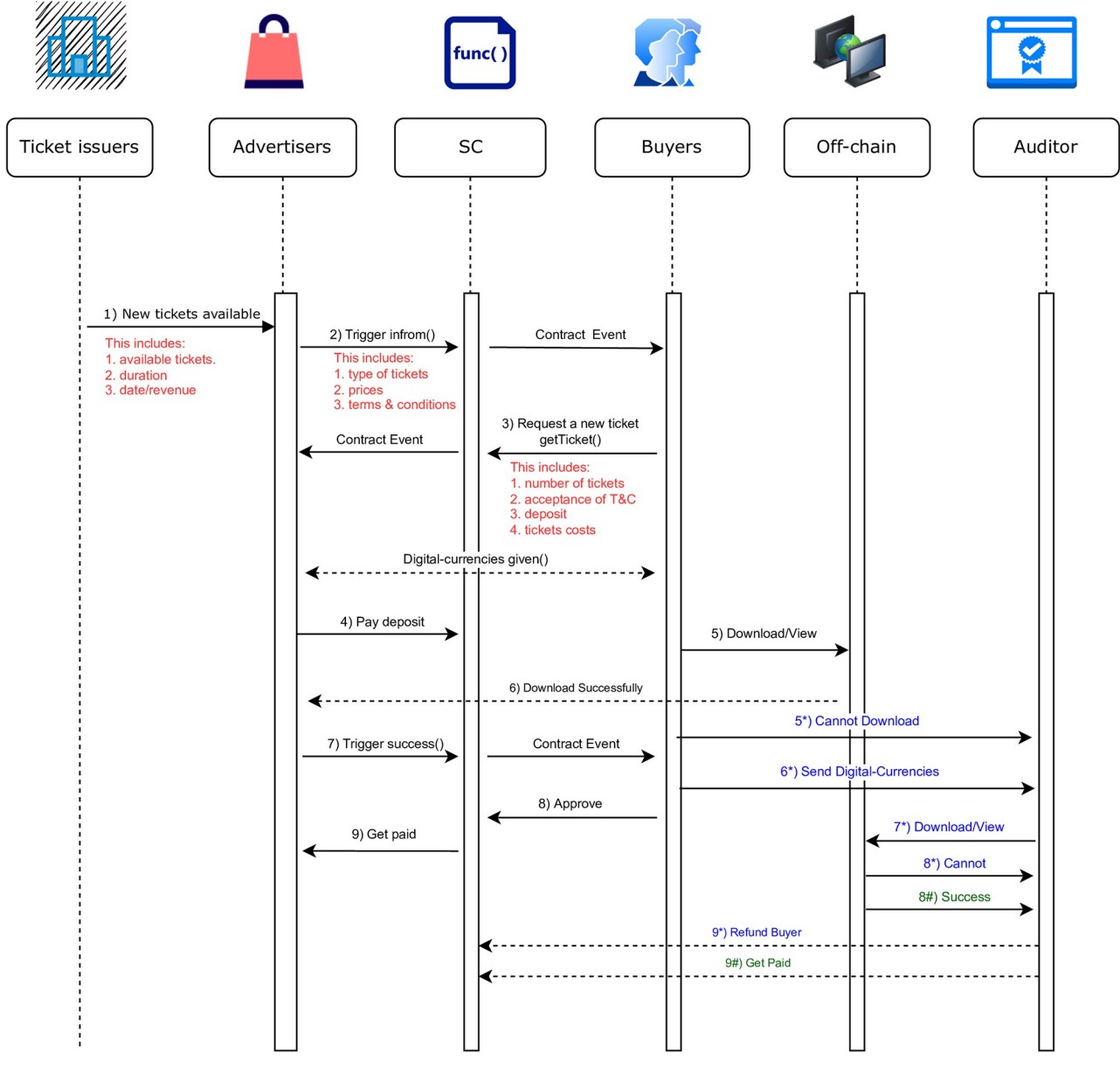

**Fig 3. BlockTicket sequence diagram.**

beginning with the distribution of tickets and ending with the final payment. Both the advertiser and the buyer verify that the download was completed successfully in the first scenario, which is depicted in the diagram as having **denoted by Black color arrow**. Choosing one of the other two scenarios, on the other hand, will result in the advertiser reporting that the buyer has downloaded the item, despite the fact that this is not actually the case. In this particular situation, the Auditor steps in, the buyer sends the same digital currency to the Auditor, and then they examine the availability to download and/or view, as depicted by the **denoted by Green color arrow** in the diagram. The remainder of this section will go into detail regarding algorithms that are employed throughout the implementation of the smart contract.

## New tickets generation/announcement

After the ticket issuers have created new electronic tickets for an event and written their terms and conditions for requesting tickets and attending the event, the tickets are distributed. The following entities are included in every event:

- Ticket ID

- Numbers of Available Tickets.

- Date and Revenue of the event.

- Price of Tickets

After these entities have been specified, ticket issuers sign each ticket with their `Private key` see Eq 1, which we assume all ticket issuers have a pair of Public and Private keys, then, issuers of tickets will notify the advertisers about the event. After that, they trigger a function of the smart contract to make an announcement regarding the occurrence. This function includes an element that maintains the particulars of the event and makes a note of the buyers see Algorithm 1.

$$SignTicket = Encrypt(TicketID, Privatekey). \tag{1}$$

**Algorithm 1:** New Tickets Announcement

```
input : EncryptedTicket, EventName, EventDate, EventLocation,NoTick-
ets,Price, Addresses, Status, T&C
output : Ready
Addresses: Array of authentic ticket issuers addresses
Access only for authentic issuers, msg.sender ∈ Addresses
if msg.sender ∉ Addresses then
  abort
else
  EvN ← EventName
  EvD ← EventDate
  EvL ← EventLocation
  nT ← NoTickets
  P ← Price
  EncTicket ← EncryptedTicket
  tc = T&C
  status ← Ready
Trigger event(Ready to process requests)
```

## Tickets requesting

Algorithm 2 demonstrates the process that the buyer goes through to request electronic tickets. Initially, the buyer will examine the terms and conditions by utilizing the `retriveTC()` function contained within the smart contrac, full implementation Available on https://github.com/mjod89/. Next, the buyer submits a request to obtain E-tickets by putting down a deposit in addition to the tickets' price. The advertiser must then make an equal deposit after this process has been completed. Later, the contract will produce its own digital currency. After the advertiser has given the client authorization to download/view their E-tickets, the customer will utilize the token, which can either be used off-chain or within the contract, to do so safely.

**Algorithm 2:** Ticket Requesting

```
input : Buyer, Deposit, Price, Status
output : Ready
Buyer: Array of authorized buyer addresses
Only authorised buyers, msg.sender ∈ Buyer
```

```
if msg.sender ∉ Addresses then
  abort
if msg.value == Deposit + Price then
  if Status == Ready then
    subtract msg.value
    generate digital currencies
    status ← ReadToDownload
  else
    Abort
else
  Abort
Trigger event (request received & Digital currencies generated)
```

## Tickets collection success/unsuccess

When the buyer downloads/view their E-tickets from the advertiser sides, in this case it could be on peer-to-peer server or on the contract itself based on the size of the tickets, which determined by the ticket issuers at the beginning, the advertiser obligated under the contract to carry out a specific function, which notifies all parties involved that the buyer has successfully downloaded/view their E-tickets.

The specifics of a successful download/view are displayed by the Algorithm 3, which then leads to an arrangement of payments but only if the buyer has given their E-tickets and is satisfied with them. As a direct consequence of this, the buyer is obligated to provide a response that attests to their level of contentment. If the buyer is content with their collection, the payment is considered to have settled, and the transaction is considered to have been successfully completed. The advertiser, as well as the buyer, would each receive the return of their deposits. In addition to this, both the issuers and the advertiser would receive a portion of the payment based on their agreements.

Nevertheless, if the buyer cannot download/view their tickets, this ultimately leads to the discontent of the buyer scenario. As a result, the only circumstance in which the auditor is requested to step in is the current one. When a buyer is displeased with their collection, Algorithm 4 explains in depth how a dispute might be addressed.

The auditor would try to retrieve the same E-tickets from the advertiser side or from the contract using the same digital currencies that was previously used. If the auditor was successful in downloading/viewing E-tickets, then the buyer's claim is unfounded, and the payment will be handled as though the transaction was successfully finished. All of the deposits are returned, and the issuers as well as the advertiser each receive compensation based on their relative agreements. However, if the auditor is unable to download/view E-tickets, the buyer will be issued a refund, and all deposits will be returned.

**Algorithm 3:** Tickets Collection success

```
input : Addresses, status, hash of Digital currencies (H),
output : Done and pay both advertisers and issuers/ Discontent
Addresses: Array of buyer and advertiser's addresses
Only authorised advertisers, msg.sender ∈ Addresses
if msg.sender ∉ Addresses then
  Abort
if H == keccak256(Digital Currencies) then
  if Status == ReadToDownload then
    Status = Confirm by advertisers
    generate an event (to buyer and issuer)
    waiting buyer to confirm
  else
    Abort
```

```
  if buyer confirm their download/view then
    return deposit
    pay costs
    status = Done
  else
    Call Auditor
    Status = Discontent
else
Abort
Trigger event (Download/view Success & Costs paid)
```

**Algorithm 4:** Tickets Collection Unsuccess

```
input : Addresses, Digital Currencies, Status
output : Issue resolved
Addresses: Array of buyer's, Advertiser's and Auditor's addresses
Only authorised Auditor, msg.sender ∈ Addresses
if msg.sender ∉ Addresses then
  abort
if Status == DiscontentCheck then
  Check ← Auditor uses digital currencies to download/view E-tickets
  if Check == valid then
    Refund buyer
    status = Resolved
  else
    pay advertiser and issuer
    status = Resolved
else
  Abort
Trigger event() (Resolved)
```

## Evaluation

An empirical analysis of the complete framework is presented, including information on how effective it is, how secure it is, how effectively it protects users' privacy, and how much it costs. This section provides an introduction to that evaluation.

## Security model

Since the suggested method is based on the digital signature cryptosystem, it will be difficult for an attacker to decrypt the signer's private key using only the signer's public key and the master domain settings. To solve this problem is similar in complexity to the RSA discrete logarithm problem. The signer's signature cannot be used by an adversary to obtain the secret key either. This is due to the fact that the signer's private key is inextricably linked to their public key via cryptography.

## Security analysis

In this section, we analyze the security benefits and drawbacks of the provided BlockTicket system.

The Ethereum Virtual Machine and the different client applications both contribute to the platform's overall security. The network will not recognize the illegal transaction until at least 50% of the nodes in the network respond in an honest manner, but a rogue node can still cause trouble by acting dishonestly. That way, the system can operate as intended. Since it is possible for a smart contract deployed inside a blockchain to have a bug or be the target of an attack, it is crucial to adhere to the security rules established by Solidity.

**Software vulnerability.** During smart contract development, we comply with the Solidity security patterns. And since the Truffle framework was explicitly made for testing smart contacts, we have put it through its paces with several different test cases, and it has always succeeded.

Furthermore, Slither [52] was fine-tuned to perform vulnerability detection with a low error rate and a minimum completion time of less than 1 second per contract. However, the complexity of the audited smart contract has a significant impact on this estimated time. A thorough audit of more complex smart contracts could take more time.

Slither can examine smart contracts written in solidity versions as low as 0.4, making it compatible with many different kinds of code. To further enhance its automation and developer friendliness, Slither may be readily linked into a CI/CD setting. Moreover, slither can unearth source code quality concerns and code optimizations that may affect gas costs.

Fig 4 shows the Slither security assessment report that was performed on our smart contract code. The code does not appear to have produced any reports of any of the security flaws or serious problems that it was intended to check for.

**BlockTicket against security features.** We outline potential attacks and risks to the entire presented framework and how those aims can be secured below. The blockchain's built-in security measures are incorporated into our system directly. Trust, integrity, non-repudiation, and availability that is not centralized are all part of these characteristics. Using the smart contract's limit modifiers, our system manages authentication and access control, ensuring that only authorized parties can carry out the contract's specified actions. Since each message exchange is cryptographically signed and time-stamped, the framework also naturally prevents Man-In-The-Middle (MITM) attacks and replay attacks. Without the legal private key, an attacker cannot forge a signature by substituting his own Ethereum address and public key for that of the original ticket holder.

**Confidentiality** Using a private or permissioned blockchain, such as the one created by Hyperledger or one of the private Ethereum blockchain networks, it is possible to fulfill the

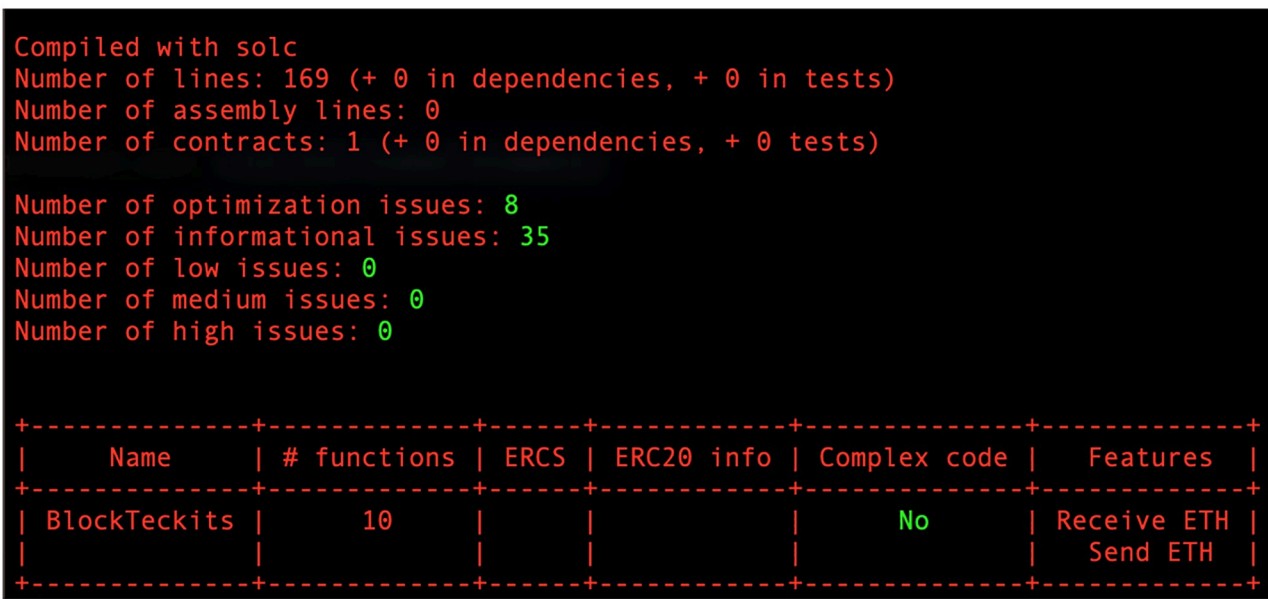

**Fig 4. Slither code analysis.**

criterion. In our case, the system is accessible to members of the general public as well as any other customer. As a result, we decided to use the public Ethereum blockchain, which is a distributed ledger that stores and transmits transactions in an open platform.

**Integrity** is an essential component that prevents any data change from taking place, which is a safeguard for essential information. Logs give users of the proposed framework the opportunity to look back through history and locate specific events. The immutability of blockchain guarantees the authenticity of all of the messages that are passed back and forth between the various participants, in addition to the logs that are created and the events that are produced. E-tickets are also held within the smart contract as well as the advertiser's side to guarantee that the one the customer downloads or views is the same as the one that is solely kept by the issuers. This is done so that the client's experience is a positive one.

**Availability** It is important to remember that once our smart contracts are put to the blockchain, they will always be accessible by the relevant parties. The availability of services is thus guaranteed at all times. Additionally, the system is safe against DoS attacks because all transactions are recorded and saved on the public Ethereum ledger in a decentralized and distributed way, making it impenetrable to hacking, failure, and tampering. Due to its decentralized, worldwide nature, the Ethereum public ledger is extremely secure and impervious to DDoS attacks. This is ensured by the tens of thousands of mining nodes that host identical records and data with a very high level of authenticity and reliability.

**Non-repudiation** Transactions on the Ethereum network both blockchain and its associated events are components of logs, in which calls are logged and the caller's identity is revealed and can't be swayed in any way There is, therefore, no way to dispute because everything they do is tracked in tamper-proof logs In addition, if an intruder tries to impersonate a legitimate buyer in a transaction between a buyer and an advertiser, they will be exposed since they lack the necessary keys.

**Privacy** Since a user's true identity is not public knowledge on the blockchain, this application allows them to buy tickets without disclosing any personal information about themselves other than their addresses. However, similar to the Bitcoin protocol, entire anonymity is not feasible within Ethereum. This is because Ethereum cannot guarantee unlinkability, and anyone with access to the blockchain could theoretically link any Ethereum account to any transaction made using that account, thereby learning at least some details about the real user's identity.

## Performance evaluation

Several evaluation metrics were utilised to determine the efficacy of the proposed framework for a secure and efficient electronic ticketing system. These evaluation measures were selected to examine the system's efficacy, reliability, and security.

A transaction's completion time was one of the metrics utilized in our study. This metric is essential since it measures the system's efficiency and the speed at which transactions may be performed. A shorter time for transaction completion would imply a more efficient system that is capable of processing transactions fast, which is essential for a ticketing system that can process a high volume of transactions.

Accuracy was another metric utilized in our evaluation. Accuracy is a crucial parameter for evaluating machine learning models, and in our instance, it indicates the system's capacity to detect fraudulent transactions with precision. A high rate of accuracy indicates that the system is effective at detecting fraudulent transactions and can lower the incidence of ticketing system fraud.

**Table 2. Performance comparison of BlockTick and [44].**

| Scheme | Availability | DoS Attack | Anonymity | Non-repudiation | Transferablility | Publicity |
|---|---|---|---|---|---|---|
| [44] | Yes | Yes | Yes | No | Yes | No |
| BlockTick | Yes | Yes | Yes | Yes | Yes | Yes |

We also examined the system's security by analyzing its resistance to assault. The threats we considered were based on the Consortium Blockchain-based Secure Electronic Ticketing System [44]. These assaults were tested against our system, which was determined to be resistant. This demonstrates that our system is secure and resilient, able to survive all forms of attacks.

To sum up, the evaluation measures we utilized were chosen to analyze the system's performance, precision, and security. These metrics were selected because they are essential for determining the efficacy of the BlockTicket, and they provide insight into the its efficacy, precision, and security. The outcomes of our evaluation indicate that BlockTicket is efficient, precise, and secure, and that it can detect and prevent fraudulent transactions with precision. It is clear from looking at Table 2 that our method is resistant to the specific attacks described in [44].

## Cost evaluation

Table 3 lists the corresponding execution and transaction costs in units of Gas. According to the table, GetTicket() has a larger execution cost than the other functions because it uses the SLOAD, SSTORE and LOG instructions to store a new request in memory and storage. These instructions utilize considerably more Gas units than a hash function. SLOAD, SSTORE and LOG use 200, 20,000 and 375 Gas units, respectively, according to the Ethereum yellow paper [40], whereas Keccak256 (hash function), which is similar to SHA3, consumes only 30 Gas units. Hence, according to the table the smart contract costs for the proposed framework seem reasonable and affordable.

## Conclusion and future work

In this article, we provide a blockchain-based solution and framework for distributing and trading of electronic ticket. Sale and distribution of electronic ticket are governed by smart contracts built on the Ethereum public blockchain. E-ticket downloads/views occur on-chain and off-chain according to the ticket size. The Ether (ETH) payments, the resolution of any disagreement that may arise, and the imposition of any necessary sanctions to encourage truthful behavior were all programmed into and thoroughly tested as part of the smart contracts. Smart contracts may make use of the decentralized peer-to-peer file system to ensure the confidentiality of the parties' agreement in its final form.

**Table 3. Cost of BlockTicket smart contract.**

| Functions | Cost in gas |
|---|---|
| ContractCreation | 2001158 |
| GetTicket() | 42090 |
| PayDeposit() | 26216 |
| refund() | 30071 |
| BuyerConfirm() | 24096 |
| ConfirmadvertiserDownload() | 30498 |
| DiscontentAndPayment() | 24432 |

In this paper, we detailed the process of creating and testing a fully functional smart contract. With the help of a security analysis, we proved that there are no exploitable security flaws in our smart contract code. We also demonstrated that our method is secure against widely used techniques.

While our proposed framework is new, it does have limits. One drawback is that the framework has only been tested on the Ethereum blockchain; additional testing on other blockchains such as Hyperledger and Avalanche are required to evaluate its performance and interoperability with other blockchain technologies. Also, the framework is confined to a specified amount of e-tickets and may not be suited for large-scale events or other sorts of digital assets other than tickets.

Another limitation is that, while the smart contracts have been extensively evaluated for security, there is always the possibility of unknown flaws or exploits, especially when new blockchain technology and attack vectors develop. Additionally, while we have explored the issue of dispute resolution in the framework of the smart contract, it may be more complex to address conflicts that are not simply quantifiable or programmatically enforceable.

We aim to overcome these limitations in future work by implementing the framework into additional blockchains, improving the functionality of smart contracts, and researching dispute resolution processes further. Overall, our architecture represents a promising approach to the distribution and trading of electronic tickets, but more research and testing are required to fully assess its potential and limitations.

## Acknowledgments

The author would like to thank the Deanship of Scientific Research at Shaqra University.

## Author Contributions

**Investigation:** Amjad Aldweesh.

**Software:** Amjad Aldweesh.

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
