## [Decision Letter · Decision Letter 0]

8 Dec 2022

PONE-D-22-31944BlockTicket: A Framework for Electronic Tickets Based on Smart ContractPLOS ONE

Dear Dr. Aldweesh,

Thank you for submitting your manuscript to PLOS ONE. After careful consideration, we feel that it has merit but does not fully meet PLOS ONE’s publication criteria as it currently stands. Therefore, we invite you to submit a revised version of the manuscript that addresses the points raised during the review process.

This paper proposes an E-ticketing framework that utilizes blockchain technology. By separating users’ credential information from financial transactions, the blockchain-based electronic ticketing model reduces the risk of data leaks and enhances privacy as well as removing third party involvement.Further there are some major parts that are missing in order to accept the paper.

1. the related work section is very weak, i recommend the author should include the tables for the related review from year 2021-22.I recommend the authors should do a critical analysis of the problem, available solution, drawbacks in current framework  for which author have written manuscript. Also used the latest research articles form 2021 & 2022 for performing related work. 

2. The abstract needs to be rewritten more precise and concrete. Including, Introduction,Objective,Method, Results, Conclusion. Please improve the abstract, it should list all the contributions made very clearly.

3  The manuscript lack in detailed description of methodology section and a flow diagram of the work. The methodology section is the core of the work, so it must be very well explained, there can be no doubts. The author must include separate methodology section and present a flow diagram of the work.

4. Manuscript lacks in comparison of the results obtained with state of art methods or models. 

5. The results section is very week to support the claim made by authors.

6. The conclusion section needs to elaborate more by discussion the disadvantages of the developed framework & discussion on the results obtain. The author should also include the future work section. 

We look forward to receiving your revised manuscript.

Kind regards,

Mohammed Shuaib

Academic Editor

PLOS ONE

5. Please remove your figures from within your manuscript file, leaving only the individual TIFF/EPS image files, uploaded separately. These will be automatically included in the reviewers’ PDF.

Additional Editor Comments:

This paper proposes an E-ticketing framework that utilizes blockchain technology. By separating users’ credential information from financial transactions, the blockchain-based electronic ticketing model reduces the risk of data leaks and enhances privacy as well as removing third party involvement.Further there are some major parts that are missing in order to accept the paper.

1. the related work section is very weak, i recommend the author should include the tables for the related review from year 2021-22.I recommend the authors should do a critical analysis of the problem, available solution, drawbacks in current framework for which author have written manuscript. Also used the latest research articles form 2021 & 2022 for performing related work.

2. The abstract needs to be rewritten more precise and concrete. Including, Introduction,Objective,Method, Results, Conclusion. Please improve the abstract, it should list all the contributions made very clearly.

3 The manuscript lack in detailed description of methodology section and a flow diagram of the work. The methodology section is the core of the work, so it must be very well explained, there can be no doubts. The author must include separate methodology section and present a flow diagram of the work.

4. Manuscript lacks in comparison of the results obtained with state of art methods or models.

5. The results section is very week to support the claim made by authors.

6. The conclusion section needs to elaborate more by discussion the disadvantages of the developed framework & discussion on the results obtain. The author should also include the future work section.

Reviewers' comments:

Reviewer's Responses to Questions

**Comments to the Author**

1. Is the manuscript technically sound, and do the data support the conclusions?

Reviewer #1: Yes

Reviewer #2: Yes

2. Has the statistical analysis been performed appropriately and rigorously? 

Reviewer #1: N/A

Reviewer #2: Yes

3. Have the authors made all data underlying the findings in their manuscript fully available?

Reviewer #1: Yes

Reviewer #2: Yes

4. Is the manuscript presented in an intelligible fashion and written in standard English?

Reviewer #1: Yes

Reviewer #2: Yes

5. Review Comments to the Author

Reviewer #1: In this research, the authors proposed an E-ticketing framework that utilizes block chain technology. By separating users’ credential information from financial transactions, the block-chain-based electronic ticketing model reduces the risk of data leaks and enhances privacy as well as removing third party involvement. The work is solid and substantive, but there are some points in the purpose for improvement.

1. In the introduction part, the strong points of this proposed article should be further stated.

2. The quality of some figures needs to be enhanced. The author(s) must redraw them with high quality. Some text on figures is difficult to read.

3. More details are needed about the experimental setup.

4. Many abbreviations are used without declaration. Abbreviated terms must be fully defined first, and then the acronyms are used.

5. The quality of the language changes depending on the sections. The authors of this article did a very good job writing it, but it could be improved.

6. The limitation and future scope of the work should be defined in the conclusion section.

7. The references are not as per the format. Some references are incomplete. Use proper complete recommended format.

Reviewer #2: The author utilizes blockchain implementation for the E-ticketing framework. The proposed approach improves throughput, decreases redundant work, and increases consensus efficiency. Also, the analysis of experimental data demonstrates the framework's benefits, which include fast ticket holding times, high throughput, and flexible scalability. The overall quality of this paper is acceptable, but I have the following comments for improvement:

1. The introduction part is not complete yet.

2. The author should elaborate more on the motivation and contribution in this part.

3. In the related work section, the authors missed many state-of-the-art works.

4. The reviewer will not name any here, but the author should carefully check and revise the work before resubmitting.

5. Presentation of the algorithm need to be improvement.

6. Presentation of the solution evaluation must be extended and much more detail

7. Conclusions must be extended and much more detail

8. English required more improvement.

6. PLOS authors have the option to publish the peer review history of their article (what does this mean?). If published, this will include your full peer review and any attached files.

Reviewer #1: No

Reviewer #2: No

---

## [Author Response · Author response to Decision Letter 0]

16 Jan 2023

Dear Editor,

We would like to thank the reviewers for their diligent and thorough reading of this paper, as well as their intelligent comments and helpful suggestions, which helped to improve its quality. We are pleased to notify you that we have updated the manuscript in response to the reviewers' recommendations. We believe that the updated manuscript is in much better shape after incorporating the feedback. We appreciate the reviewer's forthright comments: we recognized that some crucial paragraphs of the original article lacked clarity, with ambiguities that led to reader misunderstandings. We have now updated the manuscript's organization and included more paragraphs. More , we agree, clarify the description. The references have been updated in the new version. We have finally completed a thorough editing and formatting process. This update, we believe, improves the manuscript's readability. The reviewer's comments are addressed in detail below.

We hope that these revisions improve the paper such that the reviewers now deem it worthy of publication in your esteemed journal. For the reviewers' convenience, we have followed their comments by our response below. The uploaded copy of the original manuscript marked with all the changes made during the revision process. The new text is highlighted in "red colour". 

Thank you

Amjad

---

## [Decision Letter · Decision Letter 1]

13 Feb 2023

PONE-D-22-31944R1BlockTicket: A Framework for Electronic Tickets Based on Smart ContractPLOS ONE

Dear Dr. Aldweesh,

Thank you for submitting your manuscript to PLOS ONE. After careful consideration, we feel that it has merit but does not fully meet PLOS ONE’s publication criteria as it currently stands. Therefore, we invite you to submit a revised version of the manuscript that addresses the points raised during the review process.

please incorparate the comments and suggestions given by the reviewers

We look forward to receiving your revised manuscript.

Kind regards,

Mohammed Shuaib

Academic Editor

PLOS ONE

Additional Editor Comments (if provided):

please make the updation based on the comments from reviewers

Reviewers' comments:

Reviewer's Responses to Questions

**Comments to the Author**

1. If the authors have adequately addressed your comments raised in a previous round of review and you feel that this manuscript is now acceptable for publication, you may indicate that here to bypass the “Comments to the Author” section, enter your conflict of interest statement in the “Confidential to Editor” section, and submit your "Accept" recommendation.

Reviewer #3: All comments have been addressed

Reviewer #4: All comments have been addressed

Reviewer #5: (No Response)

2. Is the manuscript technically sound, and do the data support the conclusions?

Reviewer #3: Yes

Reviewer #4: Yes

Reviewer #5: Partly

3. Has the statistical analysis been performed appropriately and rigorously? 

Reviewer #3: Yes

Reviewer #4: N/A

Reviewer #5: I Don't Know

4. Have the authors made all data underlying the findings in their manuscript fully available?

Reviewer #3: Yes

Reviewer #4: Yes

Reviewer #5: Yes

5. Is the manuscript presented in an intelligible fashion and written in standard English?

Reviewer #3: Yes

Reviewer #4: Yes

Reviewer #5: No

6. Review Comments to the Author

Reviewer #3: Kudos in the good work. Nevertheless, the authors can further explain more about the data (or dataset) used (such for verification and validation) to support this research. With the dataset used for evaluation being highlighted, this would substantiate all data underlying the findings.

Reviewer #4: The paper is improved since last revision. Some minor points still need to be addressed.

In table 1, use the name of the framework instead of ours.

The visibility of Figures is low, Figures with better quality should be included.

Algorithms and figures should be referred in text close/near to Figures and Algorithms.

Reviewer #5: Overall, the idea of manuscript is average, furthermore, the contribution is not adequate at the moment. The manuscript needs significant work.

1. What are the limitations of the related works?

2. Are there any limitations of this carried out study?

3. How to select and optimize

4. the user-defined parameters in the proposed model?

5. There are quite a few abbreviations are used in the manuscript. It is suggested to use a table to host all the frequently used abbreviations with their descriptions to improve the readability

6. Explain the evaluation metrics and justify why those evaluation metrics are used?

7. Some sentences are too long to follow, it is suggested that to break them down into short but meaningful ones to make the manuscript readable.

8. The title is pretty deceptive and does not address the problem completely.

9. The related works section is very short and no benefits from it. I suggest increasing the number of studies and add a new discussion there to show the advantage. Following can be added:

10. Use Anova test to record the significant difference between performance of the proposed and existing methods.

7. PLOS authors have the option to publish the peer review history of their article (what does this mean?). If published, this will include your full peer review and any attached files.

Reviewer #3: **Yes: **Dr. Sin-Ban Ho

Reviewer #4: No

Reviewer #5: No

---

## [Author Response · Author response to Decision Letter 1]

24 Feb 2023

Dear Reviewers, 

I appreciate your valuable comments and I uploaded my responses as attached with the submission. 

Regards

---

## [Decision Letter · Decision Letter 2]

12 Mar 2023

PONE-D-22-31944R2BlockTicket: A Framework for Electronic Tickets Based on Smart ContractPLOS ONE

Dear Dr. Aldweesh,

Thank you for submitting your manuscript to PLOS ONE. After careful consideration, we feel that it has merit but does not fully meet PLOS ONE’s publication criteria as it currently stands. Therefore, we invite you to submit a revised version of the manuscript that addresses the points raised during the review process.

a few areas can be improved.

1) Introduction: The introduction should provide a clear and concise overview of the research problem, its significance, and the research questions/hypotheses addressed.

2) Methodology: The methodology should explain the data collection and analysis methods used in a clear and detailed manner, allowing for replication by other researchers.

3) Figure 1: Redraw Figure 1 to make it clearer.

4) Typos and Grammar: Address any typos or grammatical errors.. 

We look forward to receiving your revised manuscript.

Kind regards,

Mohammed Shuaib

Academic Editor

PLOS ONE

Journal Requirements:

Additional Editor Comments (if provided):

a few areas can be improved.

1) Introduction: The introduction should provide a clear and concise overview of the research problem, its significance, and the research questions/hypotheses addressed.

2) Methodology: The methodology should explain the data collection and analysis methods used in a clear and detailed manner, allowing for replication by other researchers.

3) Figure 1: Redraw Figure 1 to make it clearer.

4) Typos and Grammar: Address any typos or grammatical errors.

Reviewers' comments:

Reviewer's Responses to Questions

**Comments to the Author**

1. If the authors have adequately addressed your comments raised in a previous round of review and you feel that this manuscript is now acceptable for publication, you may indicate that here to bypass the “Comments to the Author” section, enter your conflict of interest statement in the “Confidential to Editor” section, and submit your "Accept" recommendation.

Reviewer #5: (No Response)

Reviewer #6: All comments have been addressed

2. Is the manuscript technically sound, and do the data support the conclusions?

Reviewer #5: No

Reviewer #6: Yes

3. Has the statistical analysis been performed appropriately and rigorously? 

Reviewer #5: No

Reviewer #6: Yes

4. Have the authors made all data underlying the findings in their manuscript fully available?

Reviewer #5: No

Reviewer #6: Yes

5. Is the manuscript presented in an intelligible fashion and written in standard English?

Reviewer #5: No

Reviewer #6: Yes

6. Review Comments to the Author

Reviewer #5: The quality of work is still very poor, I am not able to do further evaluation in its current form. The manuscript needs huge work to do.

Reviewer #6: This study proposes utilizing blockchain technology for E-ticketing to enhance privacy and remove third-party involvement. The experimental data highlights that this framework offers advantages such as high throughput and efficient ticket holding times. While the work is solid and substantive, a few areas can be improved.

1) Introduction: The introduction should provide a clear and concise overview of the research problem, its significance, and the research questions/hypotheses addressed.

2) Methodology: The methodology should explain the data collection and analysis methods used in a clear and detailed manner, allowing for replication by other researchers.

3) Figure 1: Redraw Figure 1 to make it clearer.

4) Typos and Grammar: Address any typos or grammatical errors.

7. PLOS authors have the option to publish the peer review history of their article (what does this mean?). If published, this will include your full peer review and any attached files.

Reviewer #5: No

Reviewer #6: **Yes: **Prof. Ammar Almomani

---

## [Author Response · Author response to Decision Letter 2]

16 Mar 2023

Dear Reviewer, 

Thank you for your comment on the importance of a clear and concise presentation in research writing and figure quality. I completely agree with you that the introduction is a critical component of any research paper as it sets the stage for the entire study as well as the methodology and the quality of figures. We have addressed all your valuable comments, please refer to the updated manuscript. 

Comment (1): Introduction: Your introduction should provide a clear and concise overview of the research problem, the significance of the problem, and the research questions or hypotheses that you are trying to address.

Reply: We would like to thank the reviewer for their valuable comment, we have improved this section kindly refer to updated manuscript line [44-52].

Comment (2): Methodology: Your methodology should describe the methods you used to collect and analyse your data. It should be clear and detailed enough to allow other researchers to replicate your study.

Reply: We appreciate the reviewer comments, we would like to inform you that the methodology section has been rewritten in an academic style to meet your valuable comment. 

Comment (3): Figure 1 show be redrawn to be clearer.

Reply: We thank the reviewer for this comment which indeed improve the final presentation of the paper, the figure has been updated. 

Comment (4): Some typos and grammar mistake that need to be addressed.

Reply: We have proofread the full paper and we believe it now looks better. Appreciate your comment. 

Regards,

Author

---

## [Decision Letter · Decision Letter 3]

27 Mar 2023

BlockTicket: A Framework for Electronic Tickets Based on Smart Contract

PONE-D-22-31944R3

Dear Dr. Aldweesh,

We’re pleased to inform you that your manuscript has been judged scientifically suitable for publication and will be formally accepted for publication once it meets all outstanding technical requirements.

Kind regards,

Mohammed Shuaib

Academic Editor

PLOS ONE

Additional Editor Comments (optional):

Dear Author,

I am pleased to inform you that your paper entitled “BlockTicket: A Framework for Electronic Tickets Based on Smart Contract” has been accepted for publication in PLOS ONE. After careful review, the reviewers and I have found your work to be of high quality and significant value to the scientific community.

Your research provides valuable insights and we believe that it will make a significant contribution to the field. We appreciate your hard work and dedication

Thank you for considering PLOS ONE as the venue for your research. We look forward to working with you in the future.

Reviewers' comments:

Reviewer's Responses to Questions

**Comments to the Author**

1. If the authors have adequately addressed your comments raised in a previous round of review and you feel that this manuscript is now acceptable for publication, you may indicate that here to bypass the “Comments to the Author” section, enter your conflict of interest statement in the “Confidential to Editor” section, and submit your "Accept" recommendation.

Reviewer #6: All comments have been addressed

2. Is the manuscript technically sound, and do the data support the conclusions?

Reviewer #6: Yes

3. Has the statistical analysis been performed appropriately and rigorously? 

Reviewer #6: Yes

4. Have the authors made all data underlying the findings in their manuscript fully available?

Reviewer #6: Yes

5. Is the manuscript presented in an intelligible fashion and written in standard English?

Reviewer #6: Yes

6. Review Comments to the Author

Reviewer #6: Dear Author,

Thank you for your prompt and thorough revisions in response to the comments provided. We appreciate your attention to detail and your efforts in addressing each of the concerns raised.

7. PLOS authors have the option to publish the peer review history of their article (what does this mean?). If published, this will include your full peer review and any attached files.

Reviewer #6: No

---

## [Editor Report · Acceptance letter]

30 Mar 2023

PONE-D-22-31944R3 

BlockTicket: A Framework for Electronic Tickets Based on Smart Contract 

Dear Dr. Aldweesh:

I'm pleased to inform you that your manuscript has been deemed suitable for publication in PLOS ONE. Congratulations! Your manuscript is now with our production department. 

Kind regards, 

on behalf of

Dr. Mohammed Shuaib 

Academic Editor

PLOS ONE